# Cardiac endothelial cells maintain open chromatin and expression of cardiomyocyte myofibrillar genes

Nora Yucel*, Jessie Axsom, Yifan Yang, Li Li, Joshua H Rhoades, Zoltan Arany*

Perelman School of Medicine, University of Pennsylvania, Philadelphia, United States

**Abstract** Endothelial cells (ECs) are widely heterogenous depending on tissue and vascular localization. Jambusaria et al. recently demonstrated that ECs in various tissues surprisingly possess mRNA signatures of their underlying parenchyma. The mechanism underlying this observation remains unexplained, and could include mRNA contamination during cell isolation, in vivo mRNA paracrine transfer from parenchymal cells to ECs, or cell-autonomous expression of these mRNAs in ECs. Here, we use a combination of bulk RNASeq, single-cell RNASeq datasets, in situ mRNA hybridization, and most importantly ATAC-Seq of FACS-isolated nuclei, to show that cardiac ECs actively express cardiomyocyte myofibril (CMF) genes and have open chromatin at CMF gene promoters. These open chromatin sites are enriched for sites targeted by cardiac transcription factors, and closed upon expansion of ECs in culture. Together, these data demonstrate unambiguously that the expression of CMF genes in ECs is cell-autonomous, and not simply a result of technical contamination or paracrine transfers of mRNAs, and indicate that local cues in the heart in vivo unexpectedly maintain fully open chromatin in ECs at genes previously thought limited to cardiomyocytes.

**\*For correspondence:**
ndyucel@pennmedicine.upenn.
edu (NY);
zarany@pennmedicine.upenn.edu
(ZA)

**Competing interests:** The authors declare that no competing interests exist.

## Introduction

Endothelial cells (ECs) are the most abundant non-blood cells in the body, and form the inner layer of all vessels in all organs. ECs carry out a wide range of critical functions, including providing a barrier between blood and the underlying parenchyma, regulating transport of nutrients and waste across that barrier, regulation of immune cell extravasation, maintaining intravascular hemostatic homeostasis, and controlling blood flow and systemic vascular resistance. These numerous roles differ widely depending on anatomical site, developmental stage, and physiological state. Consistent with this remarkable variability of function, extensive transcriptional and functional heterogeneity of ECs across tissue types, vascular localization (e.g. arterial, venous, or lymphatic), and developmental stage has been characterized in a number of studies. However, the chromatin landscape underlying these differences remains poorly understood.

Interestingly, numerous investigators have noted anecdotally, in discussions at various meetings, that cardiac ECs contain mRNAs that encode various myofibrillar proteins normally expressed in cardiomyocytes, including myosins, troponins, and titin. Most recently, Jambusaria et al. collated EC-specific RNAseq datasets from heart, brain, and lung, to formally show that ECs in these three tissues contain mRNAs normally ascribed to their respective underlying parenchyma (*Jambusaria et al., 2020*). This observation has been variably ascribed to technical artifacts during RNA preparations (i.e. contamination by RNAs from cardiomyocytes) or to presumed transfer in vivo of RNAs from cardiomyocytes to ECs via, for example, exosome transfer (although this has never been demonstrated), but these conclusions have been controversial. An alternative hypothesis is that cardiac ECs may in fact express these RNAs intrinsically. In this context, it is relevant to note

that cardiac ECs originate from tri-potential precursor cells that differentiate to cardiomyocytes, smooth muscle cells, and ECs. It may therefore be that cardiac ECs maintain epigenetic memory of a cardiomyocyte-like precursor.

To distinguish these possibilities, we use here single-cell RNAseq as well as bulk ATACSeq to characterize the chromatin accessibility signature of cardiac ECs in vivo. We show that cardiac ECs in fact maintain chromatin accessibility and transcriptional activation of myofibrillar genes typical of underlying cardiomyocyte parenchyma, but not of the immediately surrounding stromal cells. The presence of these mRNAs in cardiac ECs is thus a direct consequence of their expression in ECs.

## Results

### Cardiac ECs express myofibrillar genes normally associated with cardiomyocytes

We first sought to define the complete signature of actively translated mRNAs in cardiac ECs. To do so, we leveraged the NuTRAP mouse model, in which ribosomes are genetically tagged with GFP, conditional on Cre expression (*Roh et al., 2017*). EC-specific Cdh5-Cre NuTRAP mice thus allow for immuno-precipitation of ribosome-bound, and thus actively translated, RNA strictly from ECs, without the need for cell-isolation. RNASeq from NuTRAP-tagged cardiac ECs showed, as expected, enrichment of known EC-specific genes. Surprisingly, however, we also found significant contribution of mRNAs from cardiomyocyte myofibril (CMF) genes which, although depleted compared to the non-endothelial fraction, were still amongst the highest expressed genes in cardiac ECs (*Figure 1A*). To begin to ascertain whether this contribution was due to technical contamination, that is pulldown of mRNAs originating from cardiomyocytes instead of ECs, we performed Kendall-Tau rank-order analysis. As did Jambusaria et al, we reasoned that contamination would reflect the relative expression levels of CMF genes from the whole heart (*Figure 1A–B*). Endothelial and non-endothelial heart expression of highly expressed genes were only poorly correlated (0.216), suggesting that the presence of CMF mRNAs is not solely due to contamination.

To determine if the expression of CMF genes is enriched in ECs from the heart compared to other tissues, and to determine if other investigators had also detected CMF gene expression in heart ECs, we gathered expression data from published gene expression omnibus. We focused on datasets that used prospective isolation of ECs by flow cytometry using either genetic or surface markers in order to minimize contamination by the underlying parenchyma. All data analyzed was from uncultured, freshly isolated ECs or endothelial material. Three mouse studies (*Cleuren et al., 2019*; *Coppiello et al., 2015*; *Nolan et al., 2013*) and one study using human fetal tissue (*Marcu et al., 2018*) were identified. We first focused on the Cleuren, et al dataset, which, akin to the NuTRAP system we used above, used the *Rpl22^{fl/fl} Tek-Cre^{+/0}* mice to characterize the endothelial translatome in various tissues. Similar to our own analyses, as well those published by Jambusaria et al, we found significant expression of CMF genes in the EC translatome in this dataset (*Figure 1B*). Moreover, the expression of these genes was limited to heart ECs, and not seen in translatomes from ECs in other organs. Kendall Tau rank analysis of the expression of the top 300 highest expressed genes (which included CMF genes) in ECs vs the total heart translatome again showed low correlation (0.05–0.28, *Figure 1C*), casting doubts on the notion that CMF mRNAs in ECs originate from contaminating cardiomyocyte-derived mRNAs. To take an unbiased approach to this question, we clustered by principle component analysis the translatome RNAseq datasets, comparing various whole tissues to their respective ECs (*Figure 1D*). Multidimensional scaling of these data showed that ECs surprisingly cluster closer to their tissue of origin than to other ECs, and this was particularly true in the heart.

The next three datasets used FACS-isolated ECs to separate EC mRNAs, rather than immunoprecipitated EC-specific translatomes. Despite the different approach, analyses of these datasets again showed an enrichment of CMF gene expression in cardiac ECs, but not in ECs from other organs (*Figure 1E–F*). In contrast, expression of known cardiomyocyte-specific transcription factors, as well as metabolic genes highly enriched in cardiomyocytes, were not significantly different in cardiac ECs compared to other ECs. Together, analyses of these datasets demonstrate that CMF gene

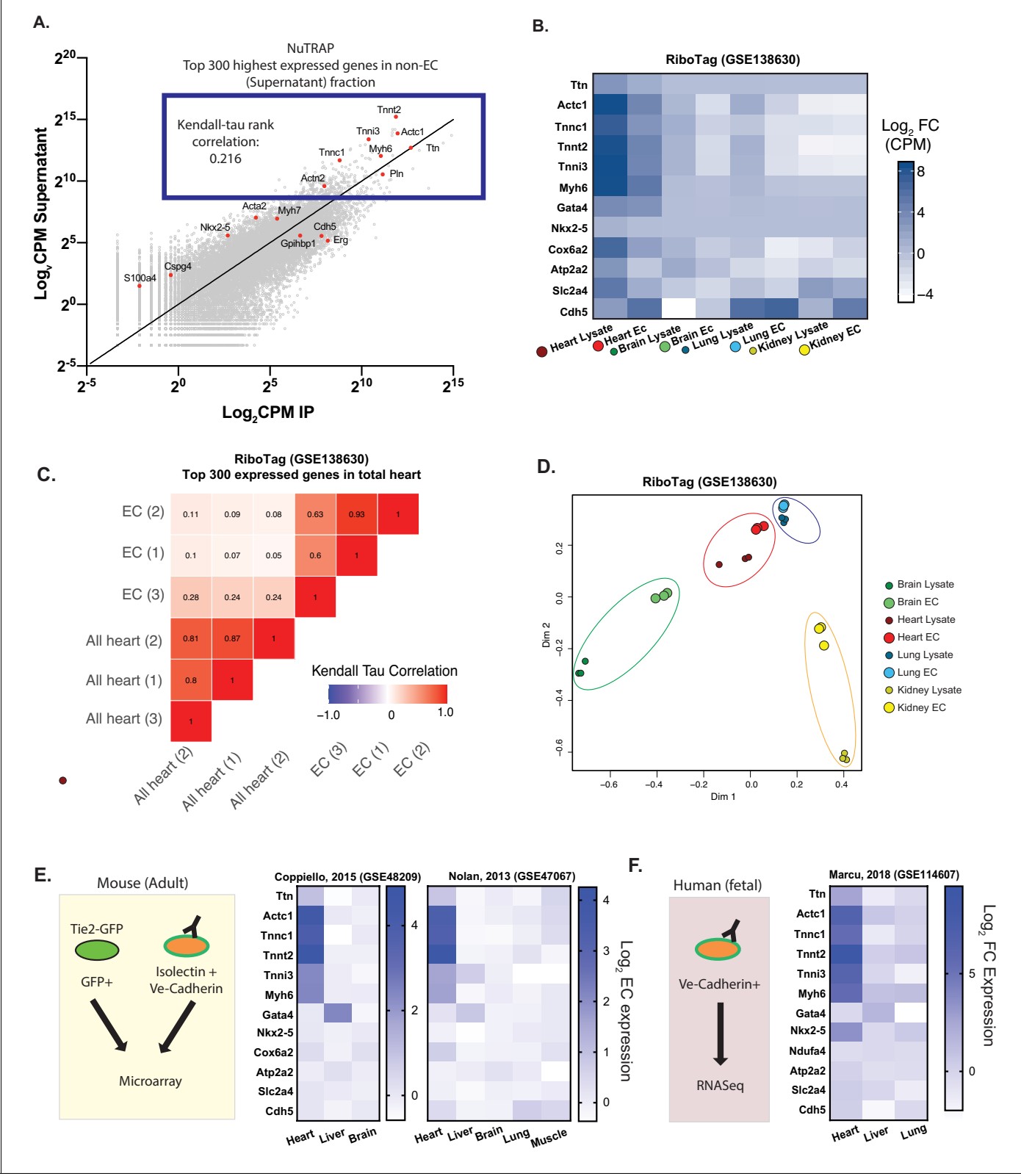

**Figure 1.** Cardiac endothelial cells express myofibrillar genes normally associated with cardiomyocytes. (**A**) RNASeq of Translating Ribosome Affinity Purification (TRAP) endothelial RNA freshly isolated from hearts of Cdh5-NuTRAP animals. Expression (Log₂CPM) of immunoprecipitated endothelial RNA (IP) versus non-endothelial RNA from the remaining fraction after IP (supernatant). Shown in the blue box are the top 300 expressed genes, with CMF genes highlighted in red. Kendall-Tau rank analyses performed on these top 300 expressed genes. (**B**) Relative expression of selected genes from

*Figure 1 continued on next page*

Figure 1 continued

freshly isolated tissue-specific adult endothelial cell TRAP, collected from (*Rpl22fl/fl*, *Tek2-Cre+*)/animals (Cleuren, et al, GSE138630) vs total tissue TRAP RNA (*Rpl22fl/fl*, *EIIa-Cre+/0*). (C) Kendall-Tau analysis of TRAP vs total tissue of heart tissue from Cleuren, et al. Analyses performed on the highest 300 expressed genes in the total heart tissue (D) Multidimensional scaling (MDS) clustering of the top 1000 variably expressed genes in the tissues analyzed in (B). (E) Relative expression of the indicated genes in tissue-specific adult murine endothelial cells in data sets from *Coppiello et al., 2015* (GSE48209) and *Nolan et al., 2013* (GSE47067) (F) Relative expression of tissue-specific fetal human endothelial RNASeq from *Marcu et al., 2018* (GSE114607).

expression in ECs: (1) is unique to cardiac ECs; (2) is generally limited to CMF genes and not to other genes highly expressed in cardiomyocytes such as mitochondrial genes; and (3) has been detected by numerous previous studies, including most recently Jambusaria et al.; and (4) is unlikely to be caused solely by contamination.

## Expression of at least one CMF gene is seen in ~60% of cardiac ECs

We next wanted to determine if CMF gene expression in ECs is ubiquitous to all cardiac ECs, or is limited to a subset of ECs. In addition, we wanted to determine whether CMF gene expression is also present in other resident cell types in the heart. To do so, we leveraged *Tabula Muris* single-cell RNAseq data (*Tabula Muris Consortium et al., 2018*), and compared CMF gene expression in cardiac ECs and fibroblasts. We first used the single-cell analysis package Seurat v3 (*Stuart et al., 2019*) to identify marker genes characteristic of cardiomyocytes, ECs or fibroblasts, that is signatures for each cell type (*Supplementary file 1*). In the cardiomyocyte signature, we selected 15 genes that were expressed in >60% of cardiomyocytes, and whose expression was at least 10-fold higher than that in non-cardiomyocytes cells. Expression of each of these 15 genes was then evaluated in single cardiac and lung ECs and fibroblasts. The number of cardiac ECs identified as expressing each of these cardiomyocyte signature genes was dramatically higher than lung ECs or fibroblasts, and 2–4 fold higher than heart fibroblasts (*Figure 2A*). A total of 59.3% of all cardiac ECs expressed at least one of the 15 CMF genes, 1.75-fold more often than cardiac fibroblasts, and about five times more often than lung ECs (which were driven primarily by one gene, *Myl4*; *Figure 2A–B*). The majority of CMF-expressing ECs only showed expression of one or two genes (*Figure 2B*). This distribution, as opposed to one subpopulation of cells expressing all CMF genes, suggests that expression is not being driven by cardiomyocytes misclassified as ECs, or by EC/cardiomyocyte doublets, again arguing against contamination.

Finally, to test in yet another way for evidence of contamination, we looked for the presence of immature, pre-spliced mRNAs of cardiac genes in ECs. We reasoned that contaminating RNAs from cardiomyocytes would originate largely from the cytosol, and therefore be primarily mature, spliced RNA. Velocyto (*La Manno et al., 2018*) was used to quantify spliced and unspliced gene counts for cardiomyocyte, EC, and fibroblast cell subsets as annotated in *Tabula Muris.* Within each subset, cells were merged, and the ratio of unspliced mRNAs/total mRNAs was calculated for each population (*Supplementary file 2*). We focused on cardiomyocyte genes with detectable unspliced mRNAs (>1%; a total of 53 genes). In this gene set, the median spliced percentages for cardiac ECs and fibroblasts cells were 5.4% and 4.2%, respectively, markedly higher than in cardiomyocytes (2.4%; *Figure 2C*). An example of this is shown in the RNA-seq track for *Myom2* in *Figure 2D*, illustrating: (1) significant expression of these genes in cardiac ECs, but not in lung ECs and (2) importantly, significant presence of unspliced introns in ECs, but not in cardiomyocytes.

Overall, these data demonstrate: (1) that EC expression of CMF genes is limited to cardiac ECs; (2) that cardiac fibroblast cells also express these genes but much less frequently than cardiac ECs; (3) that at any given time only a subset of cardiac ECs express only a subset of CMF genes, suggesting stochastic expression; and (4) that cardiac ECs express both spliced and unspliced RNAs from CMF genes, the latter at a higher frequency than cardiomyocytes, indicating cell-intrinsic expression.

## CMF mRNAs are detectable in EC nuclei ex vivo and in situ in vivo

The results above strongly suggest, but fall shy of formally proving, that the presence of CMF mRNAs in cardiac ECs is not caused by contamination from cell-free RNA derived from cardiomyocytes, whose cytoplasmic volume is much larger than fibroblasts or ECs. To address this possibility in yet another way, we isolated EC nuclei, rather than whole cells, from mouse hearts, by fluorescent sorting from Cdh5-Cre/NuTRAP animals. To confirm specificity of the sorting strategy, nuclei were

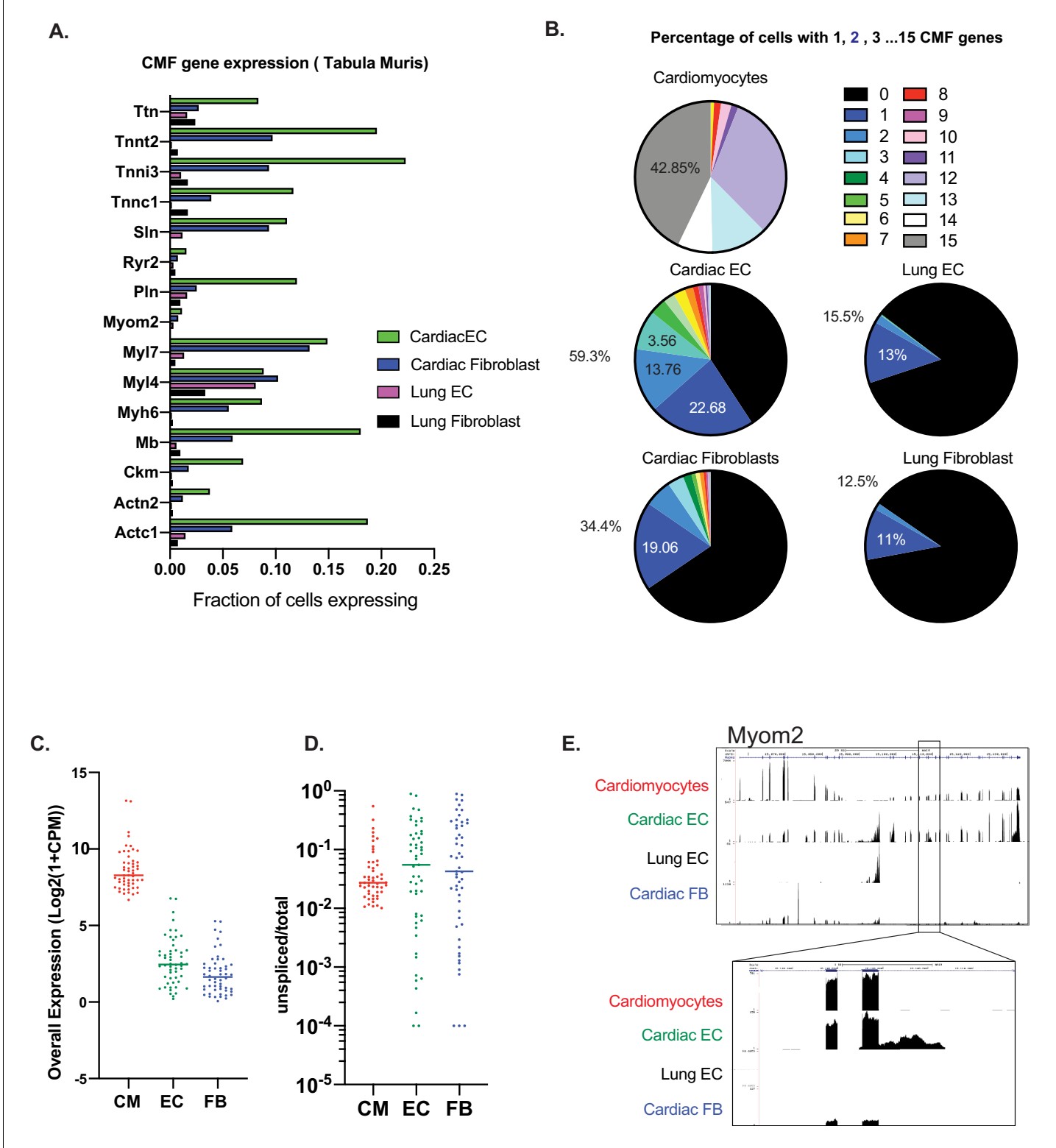

**Figure 2.** Expression of at least one CMF gene is seen in ~59% of cardiac ECs. (**A**): Analysis of mouse heart and lung single cell data from Tabula Muris in Schaum, et al (Nature, 2018). Fifteen cardiac myofibrillar (CMF) genes were chosen based on high, unique expression in the cardiomyocyte cell subsets. Positive expression was determined as a normalized count (ln(1+Counts-per-million)) greater than or equal to one. (**B**): Percentage of cardiomyocytes vs endothelial cells or fibroblasts within the heart and lung that express 1–15 of the selected CMF genes. (**C–E**). Analysis of unspliced vs spliced transcripts in *Tabula Muris* data. Data shown are merged counts for all cells within the cardiomyocyte (CM), endothelial cell (EC), or fibroblast
*Figure 2 continued on next page*

*Figure 2 continued*

(FB) cell subsets, as annotated in *Tabula Muris*. Shown are CMF genes with a unspliced percentage >= 1% in cardiomyocytes. Marker genes for cell-subsets are shown in **Supplementary file 1**, and unspliced/total ratios can be found in **Supplementary file 2** (C) Overall expression shown as log$_2$(1 +CPM) for selected CMF genes (D) Unspliced/total fraction for selected CMF genes in cardiomyocyte, endothelial cells or fibroblast cell subsets within the heart. (E) Genome track for aligned *Tabula Muris* RNA-Seq data for cardiomyocyte, cardiac EC, lung EC or cardiac fibroblast populations. In upper panel is the full track for CMF gene *Myom2,* with inset showing a regions with intronic reads specific to cardiac ECs.

co-stained with PCM-1, known to specifically mark cardiomyocyte nuclei (**Bergmann et al., 2011**), which revealed no overlap between GFP+ (EC) and PCM-1+ (cardiomyocyte) nuclei populations (**Figure 3A**). Doublets were eliminated by FSC-A/FSC-H and as well as DAPI gating (2 n) (**Figure 3— figure supplement 1**). Furthermore, the observed ~20% GFP+ (EC) nuclei and ~30% PCM-1+ (cardiomyocyte) nuclei matched previously estimated reports of cardiac cell content (**Hu et al., 2018**; **Pinto et al., 2016**). Quantitative PCR with RNA from GFP+ (EC) nuclei showed strong enrichment (~5-fold) of EC-specific transcripts (*Pecam1, Gpihbp1*), and depletion (10–15 fold) of fibroblast/pericyte-specific transcripts (*Dpt, Cspg4, Pdgfrb*). In contrast, CMF genes were depleted to a much lesser extent (2–3 fold; *Tnnt2, Myh6, Myh7, Actc1*) (**Figure 3B,C**). Hence, CMF gene expression persists in nuclei isolated from cardiac ECs.

Tissue dissociation and nuclear fluorescent sorting could, in principle, aberrantly activate expression of CMF genes in EC nuclei. To rule out this potential artifact of nuclear isolation, we used RNAscope to directly visualize CMF RNA within cardiac EC nuclei in situ in intact hearts. Using confocal imaging, we found co-localization of Tnnt2 RNA within ECs in the heart, as identified by Pecam1 (CD31) antibody staining (**Figure 3D**; **Figure 3—figure supplement 2A–B**). Quantification of nuclei with co-localization of both *Cdh5* and *Tnnt2* RNA showed presence of *Tnnt2* in ~12% of endothelial (*Cdh5*+) cell nuclei (**Figure 3—figure supplement 2C–D**), a proportion that is similar to the number of ECs in Tabula Muris that are Tnnt2 positive (**Figure 2A**). Altogether these results unambiguously show that expression of CMF genes in ECs is not due to contamination from parenchymal cells, and strongly suggest that expression of CMF genes stems from direct expression from EC chromatin.

## Cardiac ECs maintain open chromatin at CMF genes

The presence of CMF transcripts in ECs, if not technical contamination, could potentially reflect a biologic paracrine contribution of cardiomyocyte-derived RNAs transmitted to ECs via, for example, exosomes or microvesicles, as previously described in non-cardiac contexts (**Skog et al., 2008**; **Valadi et al., 2007**). On the other hand, our single-cell analysis, RNAScope and nuclear sorting data strongly suggested EC-intrinsic transcription of CMF genes. To distinguish these possibilities, we used ATAC-Seq to determine directly whether cardiac ECs have open chromatin at CMF genes, reasoning that if there is EC-intrinsic expression, there should also be some detectable open chromatin at cardiomyocyte-specific genes. We performed genome-wide ATAC-Seq (**Buenrostro et al., 2015**) on nuclei sorted from Cdh5-Cre/NuTRAP animals, and compared GFP positive (endothelial) and GFP negative (non-endothelial) populations (**Figure 4A–C**). We then quantified average enrichment at the transcriptional start sites (TSSs) of endothelial, fibroblast, and cardiomyocyte marker gene sets (**Figure 4B**), as well as per-gene comparison of read density at the TSSs of GFP+ (EC) vs GFP- (non-EC) samples (**Figure 4C**). As expected, EC nuclei had an open chromatin signature at EC-specific genes (**Figure 4A**: *Erg, Cdh5, Gphibp1, Egfl7*; **Figure 4B** green line; and **Figure 4C**, green circles), and markedly reduced open chromatin signature at fibroblast genes (**Figure 4A**: *Dpt, Cspg4, Pdgfrb, Col6a2*; **Figure 4B** blue line; and **Figure 4C** blue triangles). Strikingly, however, the chromatin of CMF genes was quantitatively as open in ECs as in cardiomyocyte nuclei (**Figure 4A**: *Tnni3, Tnnt2, Tnnc1, Myh6*; **Figure 4B** red line; and **Figure 4C**, red diamonds). In addition to accessibility at promoter-TSS regions, most peaks associated with enhancer regions of cardiomyocyte marker genes (intergenic and intronic) were also seen in cardiac ECs (**Figure 4—figure supplement 1A–B**) We did identify a small number of enhancer regions that were unique to the non-EC nuclei, including an enhancer region upstream of the gene encoding the cardiac transcription factor *Nkx2-5*, and an intronic enhancer within the *Dmd* gene (**Figure 4—figure supplement 1C**). These data demonstrate that cardiac ECs have open chromatin at CMF genes, and exhibit quantitatively similar accessibility at these genes at both TSS and enhancer regions as the underlying parenchyma.

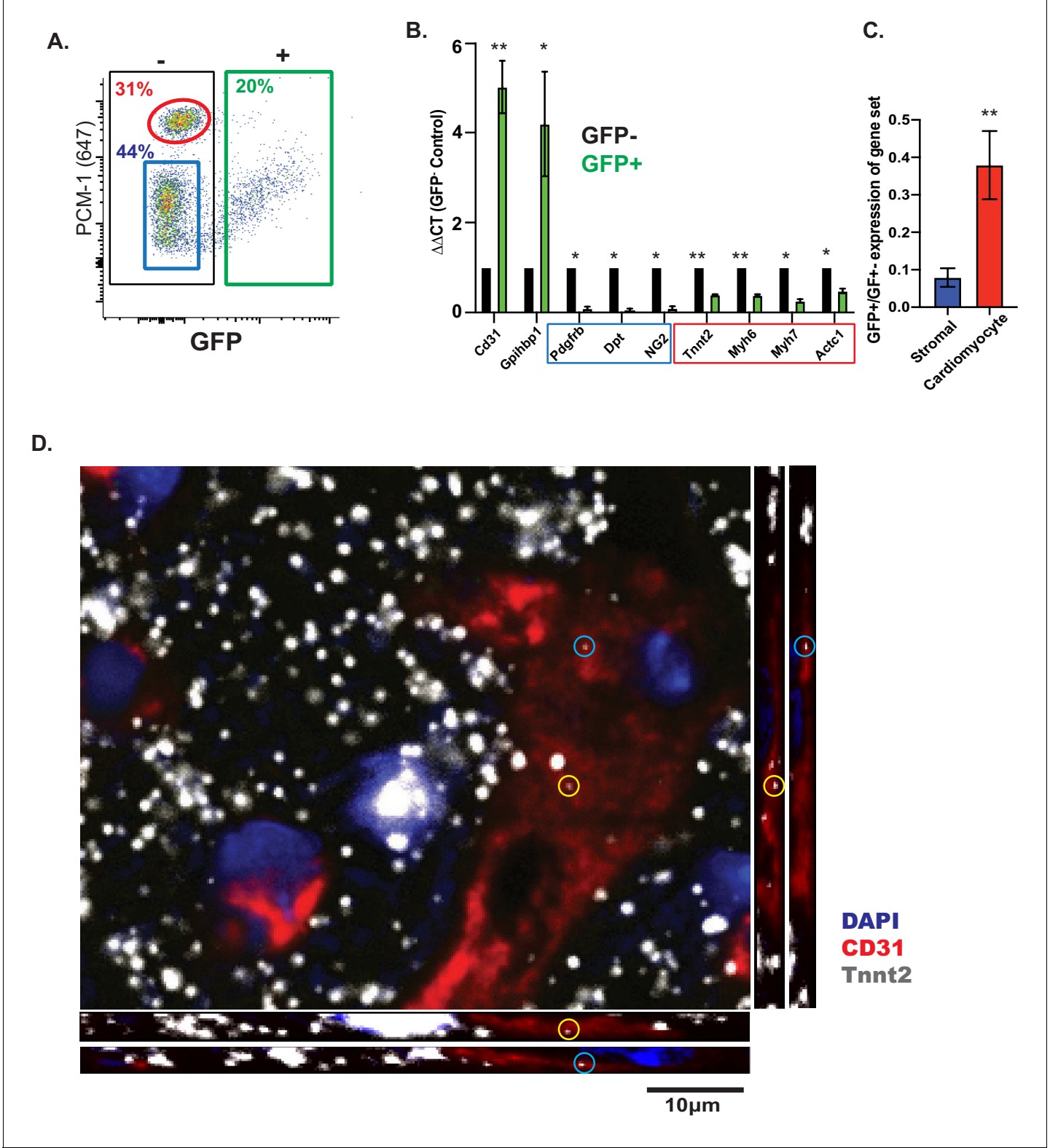

**Figure 3.** CMF mRNAs are detectable in EC nuclei ex vivo and in situ in vivo. (**A**) Representative flow cytometry plot of GFP expression from isolated nuclei from Cdh5/NuTRAP mice. About 20% of nuclei are GFP+ (endothelial), with no co-staining with the cardiomyocyte-specific nuclear marker PCM1. Gating scheme for identification of single nuclei shown in *Figure 3—figure supplement 1* (**B**) Relative expression (by qPCR) of nuclear RNA from isolated GFP+ (endothelial) or GFP- (non-endothelial) nuclei. Blue box: stromal genes; red box: CMF genes. (**C**) Average fold change of expression (GFP+/GFP-) in (**B**) for fibroblast or CMF genes. (**D**) Confocal images of heart sections. RNAScope probes for *Cdh5* (endothelial) and CMF (*Tnnt2*)

*Figure 3 continued on next page*

*Figure 3 continued*

mRNA were co-stained with Pecam1 antibody. Shown are confocal slices of 0.96 µm thickness. Additional images are available *Figure 3—figure supplement 2*.

The online version of this article includes the following figure supplement(s) for figure 3:

**Figure supplement 1.** Cdh5-Cre/NuTRAP nuclei sorting scheme.
**Figure supplement 2.** Confocal imaging of Tnnt2 mRNA by RNAScope in heart sections.

HOMER analysis (*Heinz et al., 2010*) revealed that ATAC-seq peaks unique to GFP+ (EC) nuclei (*Supplementary file 3*) were enriched for known endothelial transcription factor binding sites (ERG, ETS), consistent with their endothelial identity. Genes within 2 kb of GFP+ unique peaks showed GO enrichment for vascular development-associated terms (*Figure 4D*, lower panel). In parallel, peaks unique to GFP- (non-EC) nuclei were enriched for the smooth muscle and cardiac fibroblast transcription factor TCF21, as well as genes associated with neurogenesis and extracellular matrix organization (upper panel). Most interestingly, however, peaks that overlapped between GFP- and GFP+ nuclei (middle panel) were significantly associated with genes of cardiomyocyte development and contraction, and HOMER analysis showed enrichment for the cardiac transcription factors GATA4 and MEF2. These unbiased analyses thus strongly support genome-wide open chromatin at CMF genes in both ECs and cardiomyocytes.

Finally, we compared these epigenetic data with heart EC-specific genes (as opposed to ECs from other organs) identified in previous studies. We focused on genes with >2 fold increased expression compared to brain or lung ECs in the data from Cleuren et al., and a baseline expression of at least 10 transcripts per million (TPM) in both that data set and ours (*Figure 5A*). Interestingly, not all of these cardiac EC-enriched genes revealed open chromatin (RPKM <1.5 at the TSS), suggesting that these transcripts may in fact have origins outside the ECs. GO analysis of the genes with closed chromatin revealed largely fibroblast-like annotations (*Figure 5B*, left). In contrast, GO analysis of the genes with open chromatin (RPKM >3 at the TSS) once again revealed strong enrichment for CMF genes, as well as metabolic genes related to handling of fatty acids, the primary fuel of cardiomyocytes (*Figure 5B*, right).

In summary, these data demonstrate that, while as expected cardiac ECs have open chromatin at EC-specific genes and closed chromatin at fibroblast cell-specific genes, they broadly share open chromatin regions with the cardiomyocytes that make up the underlying parenchyma.

## Open chromatin and expression of CMF genes in ECs requires in vivo cues

Finally, to determine whether open chromatin at CMF is an indelible property of cardiac ECs, or whether it requires active maintenance, we isolated primary cardiac or lung ECs and assayed open chromatin by ATAC-qPCR and ATAC-Seq after expansion in culture, that is in the absence of local cues provided in vivo (*Figure 6A*). Chromatin at the EC-specific genes *Erg* and *Cdh5* remained open during this time, consistent with maintenance of endothelial identity (*Figure 6B*). In sharp contrast, chromatin at CMF genes (*Tnnt2, Myh6*) completely closed within this time span, and had similar accessibility to freshly isolated lung endothelial nuclei (*Figure 6C*). Interestingly, accessibility at *Pdgfrb*, a pericyte/fibroblast marker, was increased in culture, although *Cspg4* and *Tcf21* remained closed (*Figure 6D*), likely reflecting some endothelial-to-mesenchymal transition occurring under culture conditions. Similarly, ATAC-Seq of cultured cardiac ECs showed that accessibility at CMF genes was overall reduced compared to endothelial or fibroblast genes (*Figure 6E*). Annotation of ATAC-Seq peaks (*Figure 6F*; *Supplementary file 4*) showed an enrichment for in genes involved with metabolic processes, signal transduction, and cell differentiation in newly opened peaks. In contrast, peaks that closed in culture were primarily associated with CMF terms, including heart development and muscle contraction. We conclude that open chromatin at CMF genes in cardiac ECs in vivo requires active maintenance by the surrounding parenchyma.

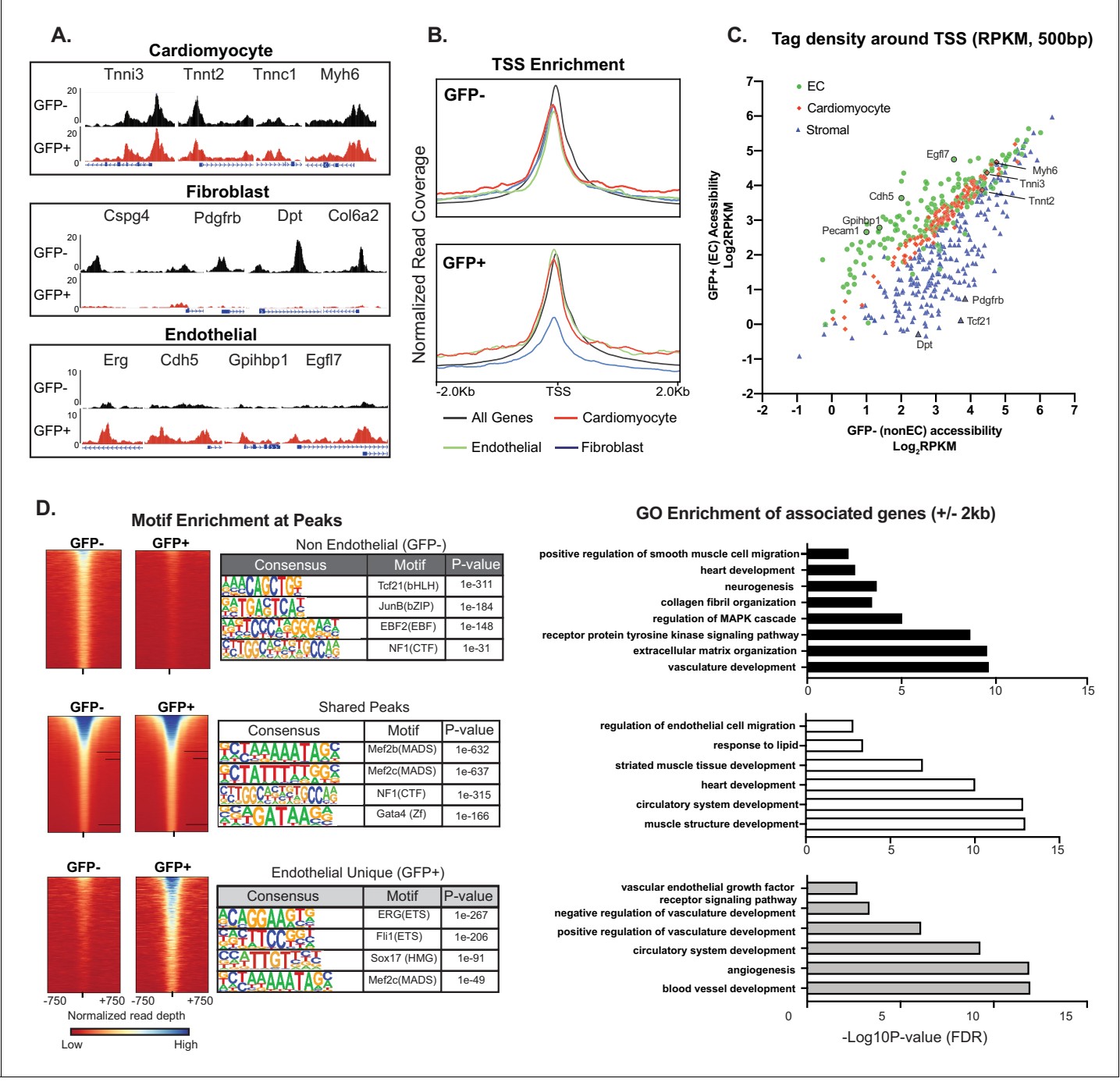

**Figure 4.** Cardiac ECs maintain open chromatin at CMF genes. (**A**) Representative ATACSeq gene tracks for cardiomyocyte, fibroblast, or endothelial cell genes in isolated GFP+ (endothelial, in red) or GFP- (non-endothelial, in black) nuclei. (**B**) Genome-wide open chromatin at the transcriptional start sites (TSSs) of cardiomyocyte, stromal, or endothelial cell genes in GFP- (non-endothelial) and GFP+ (endothelial) nuclei. Note chromatin of cardiomyocyte genes (red) is as open in ECs (GFP+) as non-ECs (GFP-). (**C**) Comparison of accessibility in GFP- vs GFP+ nuclei at TSS peaks (+/- 250 bp) for EC genes (green), cardiomyocyte genes (red), and stromal genes (blue). Note again that chromatin of cardiomyocyte genes is as open in EC as non-EC nuclei. (**D**) Motif enrichment analysis of ATACSeq peaks unique to non-endothelial (GFP-) nuclei (top), unique to endothelial (GFP+) nuclei (bottom), and shared peaks (middle). Right panels: gene ontology (GO) analysis of genes within 2 kb of each peak set. Full statistics and GO annotations for peak regions shown in *Supplementary file 3*. Additional analyses shown in *Figure 4—figure supplement 1*.

The online version of this article includes the following figure supplement(s) for figure 4:

**Figure supplement 1.** Differential accessibility at all peaks associated with CMF genes.

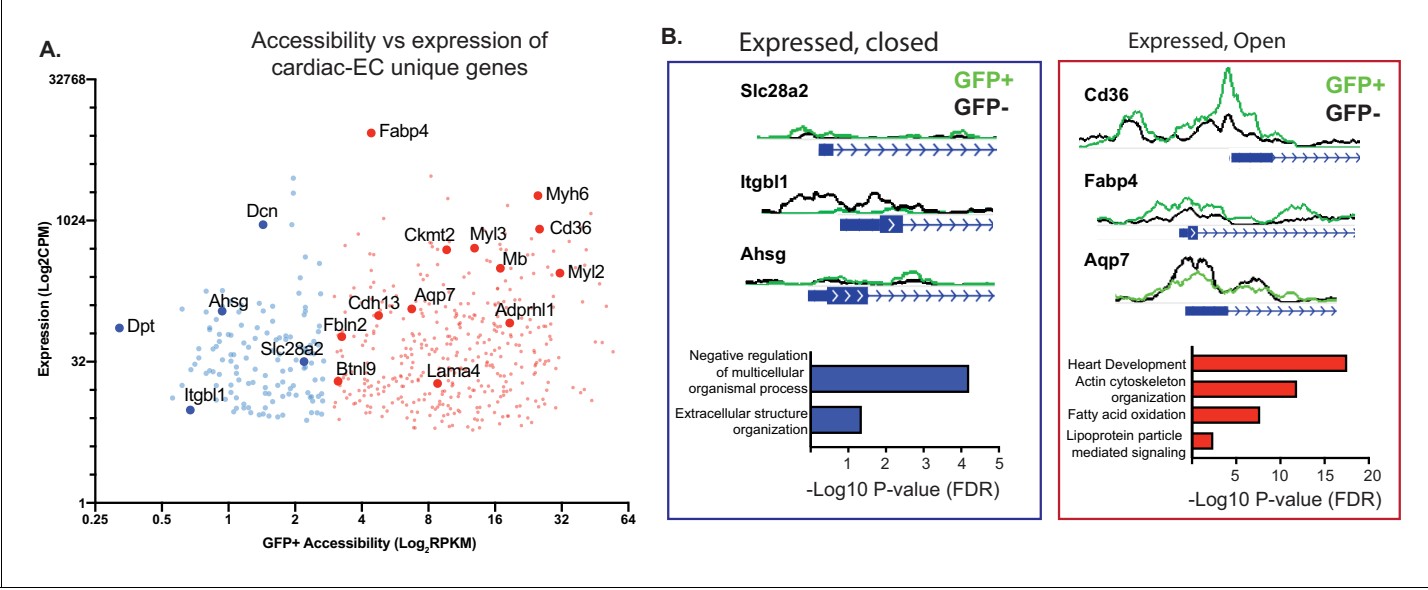

**Figure 5.** Not all genes expressed in endothelial cells (ECs) have open chromatin. (**A**) Gene expression (data from Cleuren et al.; GSE138630) and gene accessibility of cardiac EC-enriched genes (with expression of at least 10 CPM, and twofold or higher expression compared with brain or lung endothelial cells) in cardiac endothelial (GFP+) nuclei. Blue: relatively close chromatin (Log2RPKM +/- 250 bp at TSS peaks <3); red: relatively open chromatin (>3). (**B**) Representative tracks and gene ontology (GO) analysis of cardiac EC-enriched genes with closed (blue) or open (red) chromatin.

## Discussion

Overall, we show here that cardiac ECs possess transcriptional and epigenetic signatures of cardio-myocytes, but not other cell types within the heart. Moreover, the presence of open chromatin at cardiomyocyte-specific genes, as well as the nuclear expression of these genes in ECs, demonstrates significant EC-intrinsic transcriptional contribution of these mRNAs. A gene expression signature in ECs that mirrors the underlying parenchyma is also apparent in public datasets from other studies, and was recently reported in the literature for the first time by Jambusaria et al. These studies, however, left open the question of the mechanism by which these mRNAs were to be found in cardiac ECs. In numerous unofficial discussions at scientific meetings, the observation has in general been ascribed to either technical contamination, or to cardiomyocyte-derived mRNA transferring to ECs via extracellular vesicles or other paracrine mechanism. Our results, in contrast, demonstrate that these CMF mRNAs originate in ECs themselves, via active maintenance of open chromatin and transcription. It must be noted that the functional consequence of this shared signature, and whether it results in translation of CMF transcripts, remains to be determined.

The notion that ECs actively express CMF genes has been contentious. However, at this point, the arguments against contamination as an explanation for the presence of CMF mRNAs in cardiac ECs are numerous and compelling:

1. Kendall Tau correlations demonstrate lack of correlation between genes highly expressed in cardiomyocytes and those potentially contaminating ECs. Were contamination to be occurring, these levels of expression would correlate.
2. In the single cell RNAseq datasets, only a subset of ECs contained only a subset of CMF genes (most often only one). If CMF mRNAs were being captured by ECs by contamination, this would be observed in all cells. Even in the unlikely scenario that only a subset of ECs captured CMF mRNAs, they would capture all CMF genes instead of on average only one.
3. CMF mRNAs in ECs are relatively more unspliced than in cardiomyocytes, supporting the presence of nascent mRNAs in ECs.
4. Nuclear qPCR reveals the same observations as whole-cell RNAseq. If contamination were occurring in this setting, it would require the formation of nuclear doublets, which was ruled out by forward/side scatter, and by counterstaining with PCM-1, a marker specific for cardiomyocyte nuclei.

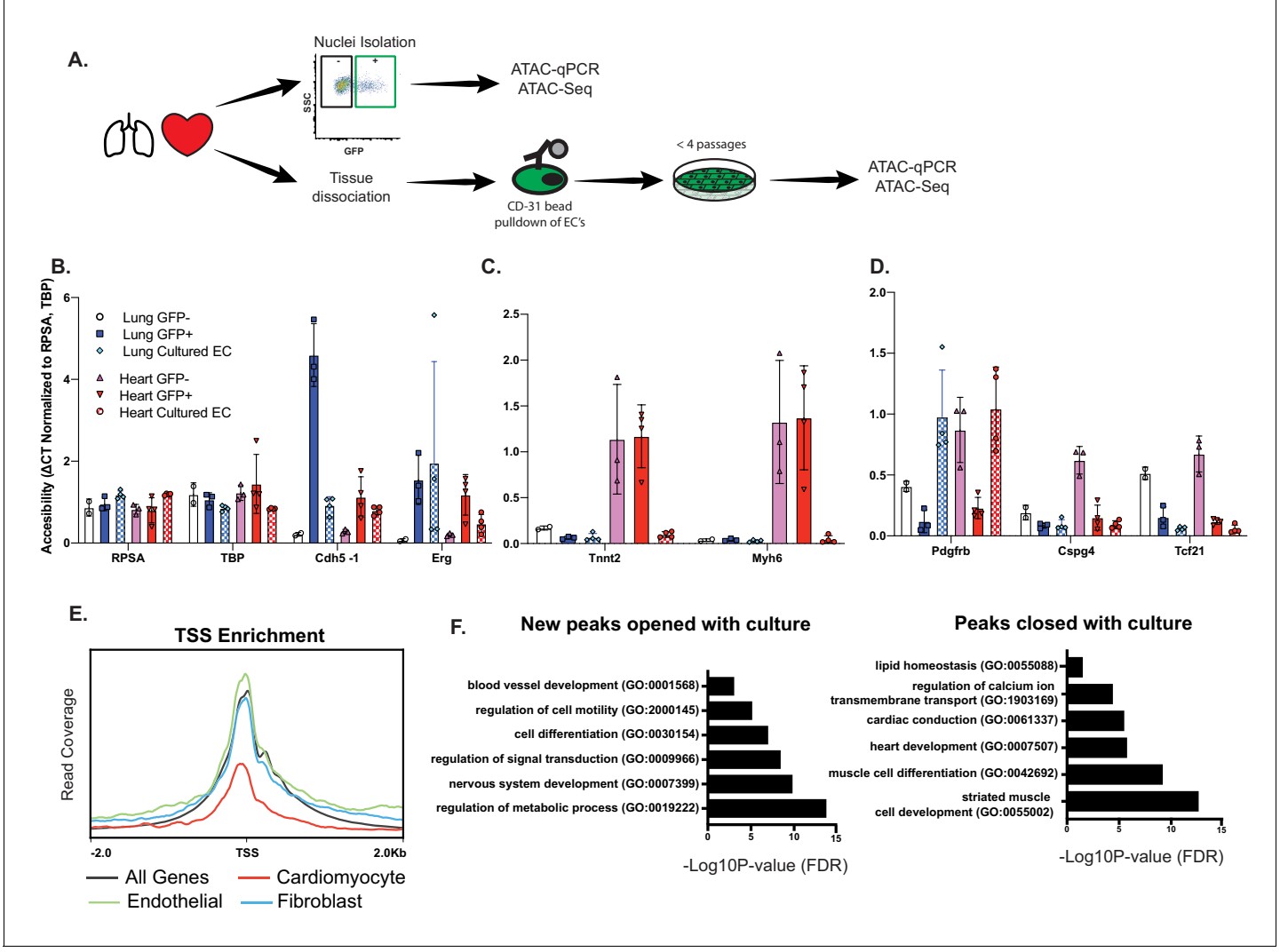

**Figure 6.** Open chromatin and expression of CMF genes in ECs requires in vivo cues. (A) Experimental scheme. N = 2–4 biological replicates. ATAC-qPCR and ATAC-Seq was performed on freshly isolated endothelial nuclei, or cultured endothelial cells from heart or lung. qPCR regions were identified based on peaks identified in sequencing data. (B) Relative accessibility (compared to housekeeping genes, *Rpsa* and *Tbp* of endothelial cell genes, *Cdh5* and *Erg*). (C) Relative accessibility of cardiomyocyte genes (*Tnnt2, Myh6*). (D) Relative accessibility of fibroblast-signature genes (Pdgfrb, Cspg4, Tcf21) (E) Representative TSS enrichment plot of cultured cardiac endothelial cells. Enrichment shown for cardiomyocyte-specific, endothelial or fibroblast specific gene sets (gene sets previously identified in *Supplementary file 1*). (F) GO enrichment for peaks that open (left) or close (right) in cultured cardiac endothelial cells vs freshly isolated. Complete statistical analyses by DESEQ2 and GO annotations shown in *Supplementary file 4*.

5. RNAscope studies demonstrate direct visualization of CMF transcripts in EC nuclei in situ, ruling out artifacts of cell preparation or sorting during the RNAseq studies.

6. ATACseq on chromatin of cardiac ECs demonstrates fully open chromatin at CMF genes. Unlike highly variable mRNA abundance, chromatin abundance is proportional to cell count. Thus, if contamination were occurring in this setting, EC chromatin would be, on average, only mildly open (proportional to the level of contamination), because the majority of the signal would still stem from ECs.

7. In aggregate, these data provide unambiguous support for the notion that cardiac ECs have open chromatin at CMF genes, and actively transcribe these genes. However, these data do not indicate whether CMF genes are also translated, and serve similar functions in ECs as they do in cardiomyocytes.

Interestingly, we find that the expression of CMF genes, including immature, unspliced transcripts, in ECs is detectable in only a subset of cells (~60%), and that in general only 1–2 genes are

detectable, and not always the same ones. In contrast, the chromatin at these genes, although evaluated only in bulk, appears to be as open as that of cardiomyocytes. These observations suggest that chromatin is maintained open at these genes in all cardiac ECs, but that rates of transcription are relatively low and stochastic, and that there may not be a defined subset of ECs that express CMF genes. We hypothesize that this fully open chromatin at CMF genes in cardiac ECs may reflect shared developmental origin between cardiomyocytes and ECs. During development, cardiac progenitors that express *Nkx2-5* (primary heart field) or *Isl1* (secondary heart field) give rise to both ECs and cardiomyocytes (*Jia et al., 2018*; *Moretti et al., 2006*; *Wu et al., 2006*). Thus, accessibility of chromatin at cardiomyocyte-specific genes may be due to epigenetic memory of this shared developmental origin.

An outstanding question remains as to the role of the cardiomyocyte epigenetic and transcriptional signature on endothelial function. Recent studies point towards a potential role in EC maturation. Both developmental and adult cardiomyocyte and ECs share expression of the transcription factor MEF2C, the number one predicted binding site in our analysis of open chromatin peaks shared between cardiac ECs and cardiomyocytes. Endothelial-specific MEF2C has been shown to regulate angiogenesis (*Sacilotto et al., 2016*), cell integrity and survival (*Potthoff and Olson, 2007*), as well as response to inflammation (*Xu et al., 2015*). GATA4, whose motif is highly enriched between cardiac ECs and cardiomyocytes, is expressed by cardiac progenitors and in adult cardiomyocytes, but almost undetectable in ECs. Nonetheless, a recent study *Maliken et al., 2018* demonstrated that inducible knockdown of GATA4 in adult ECs resulted in a less mature, PECAM1-'low' EC phenotype, including hyperproliferation, reduced differentiation, and impaired tube formation capacity. Although not highlighted in the main figures, the supplementary data from this study reveals that expression of CMF genes, including *Myh6*, *Myl7*, *Tnnt2* and *Sln*, are significantly downregulated (25–1.5 fold) upon *Gata4* knockdown. These studies, which show that loss of cardiomyocyte transcription factors results in EC 'immaturity', are consistent with our observation that cardiomyocyte-specific gene accessibility is lost in cardiac ECs upon culture. We hypothesize that perhaps cardiomyocyte transcription factors, such as GATA4 and MEF2, are responsible for maintaining the open chromatin signature of CMF genes within ECs in the heart, and that CMF gene expression plays a role in cardiac EC maturity.

In summary, we demonstrate here that cardiac ECs maintain open chromatin and active transcription at a number of myofibrillar genes thought to be uniquely expressed in cardiomyocytes. This shared chromatin accessibility landscape is likely maintained by paracrine cues in vivo, and likely serves to maintain the unique phenotype of cardiac ECs.

# Materials and methods

## Key resources table

| Reagent type (species) or resource | Designation | Source or reference | Identifiers | Additional information |
|---|---|---|---|---|
| Strain, strain background (*Mus musculus*) | B6;129S6-*Gt (ROSA)26Sortm2 (CAG-NuTRAP) Evdr*/J | Cell Rep. 2017 Jan 24; 18(4): 1048–1061. | NuTRAP | https://www.jax.org/strain/029899 |
| Strain, strain background (*Mus musculus*) | B6;129-Tg(Cdh5-cre)1Spe/J | JAX strain 017968 | Cdh5-Cre, VE-Cadherin-CRE | |
| Antibody | Pecam1 antibody; Rat monclonal | BD Pharmigen | Cat# 558736 | (1:500) |
| Antibody | Pcm1; Rabbit polyclonal | Sigma | Cat# HPA023370 | (1:250) |
| Software, algorithm | fastp | *Chen et al., 2018* | | |
| Software, algorithm | Bowtie2 | *Langmead and Salzberg, 2012* | | |

*Continued on next page*

*Continued*

| Reagent type (species) or resource | Designation | Source or reference | Identifiers | Additional information |
|---|---|---|---|---|
| Software, algorithm | Genrich | https://github.com/jsh58/Genrich | | |
| Software, algorithm | Picard Tools | http://broadinstitute.github.io/picard/ | | |
| Software, algorithm | samtools | *Li et al., 2009* | | |
| Software, algorithm | STAR | Dobin, 2013 | | |
| Software, algorithm | Seurat | *Stuart et al., 2019* | | |
| Software, algorithm | R | https://www.r-project.org/ | | |
| Software, algorithm | DiffBind | *Ross-Innes et al., 2012* | | |
| Software, algorithm | Deeptools | *Ramírez et al., 2016* | | |
| Software, algorithm | Homer | *Heinz et al., 2010* | | |
| Software, algorithm | Velocyto | *La Manno et al., 2018* | | |
| Other | DAPI stain | Molecular Probes | | (1:1000) |
| Other | RNAscope Probe- Mm-Tnnt2-C3 | ACD Bio | Cat# 418681-C3 | |
| Other | RNAscope Probe- Mm-Cdh5-C2 | ACDBio | Cat# 312531-C2 | |
| Commercial assay or kit | RNAscope Multiplex Fluorescent v2 Assay | ACD Bio | Cat# 323136 | |
| Commercial assay or kit | Illumina Tagment DNA Enzyme and Buffer | Illumina | Cat#:20034197 | 1.25 uL of enzyme used per 50,000 nuclei for transposition |

## Nuclei isolation

Nuclei were isolated from whole heart tissue following the method outlined in *Roh et al., 2017*. In brief, whole hearts were dounced in 10 mL nuclear preparation buffer (NPB; 10 mM HEPES (pH7.5, 1.5 mM MgCl, 10 mM KCl, 250 mM Sucrose, 0.1% IGEPAL-630/NP-40, 0.2 mM DTT, cOmplete mini EDTA-free protease inhibitor Roche #11836170001)). Homogenates were filtered through 40 μM cell strainers, and spun at 500G for 5 min. Nuclear pellets were washed 1X in NPB buffer, and then resuspended in 2 mL nuclear sort buffer (NSB; 10 mM Tris pH 7.5, 40 mM NaCl, 90 mM KCl, 2 mM EDTA, 0.5 mM EGTA, 0.1% IGEPAL-630/NP-40, 0.2 mM DTT, cOmplete mini EDTA-free protease inhibitor) and passed through a filter-top FACS tube.

## TRAP RNA isolation

Translating Ribosome Affinity Purification (TRAP) was performed as described in Roh, et al. In brief, ~50 mg of fresh heart tissue was dounce homogenized in 6 mL of homogenization buffer (50 mM Tris pH7.5, 12 mM MGCl,, 100 mM KCl, 1% IGEPAL-630/NP-40, 100 μg/mL cycloheximide, 1 mg/mL sodium heparin, 2 mM DTT, 0.2units/μL RNAse inhibitor, 1X cOmplate EDTA-free protease Inhibitor). Samples were centrifuged at 13,000 RPM for 10 min to remove cell debris. The supernatant was then collected and incubated with anti-GFP antibody (5 μg/mL, Abcam ab290) for 1 hr at 4°

C. Protein G dynabeads were washed twice in low-salt wash buffer (50 mM Tris, pH 7.5; 12 mM MgCl2; 100 mM KCl; 1% NP-40; 100 µg/ml cycloheximide; 2 mM DTT), added to the homogenates with antibody, and subsequently incubated for 30 min. Dynabeads with immunoprecipitates were washed three times in high-salt wash buffer (50 mM Tris, pH 7.5; 12 mM MgCl2; 300 mM KCl; 1% NP-40; 100 µg/ml cycloheximide; 2 mM DTT). Following the last wash, RLT buffer with β-mercaptoethanol was added to dynabeads, and RNA was extracted using Qiagen Micro RNeasy kit according to the manufacturer's instructions. For input or supernatant RNA, 5% of total homogenates or IP supernatants unbound to beads (respectively) were mixed with TRIzol and processed according to the manufacturer's instructions to extract total RNA. All RNA was further cleaned using AMPure bead purification to remove residual salts, and RNA integrity was analyzed by Agilent Bioanalyzer.

## Flow cytometry and sorting

EC nuclei were isolated by florescence activated cell sorting (FACs) on a BD FACS Aria II. Single, non-dividing nuclei were identified by selecting on DAPI for 2 n, followed by FSC-A/FSC-H for doublet removal, and FSC/SSC. From this subset, GFP+ nuclei were selected. GFP positive and negative nuclei were sorted directly into ATAC-Seq lysis buffer (Buenrostro, et al; 10 mM TrisCl, 10 mM NaCl, 3 mM MgCl, 0.1% IGEPAl-630/NP-40) in eppendorf tubes for further processing.

## ATAC-Seq

Sorted nuclei were spun at 500G at 4C for 10 min. ATACSeq reactions were prepared from nuclei according to Buenrostro, et al, using Illumina TDE1 tagment DNA Enzyme and TD buffer (#15027865, 15027866). Libraries were amplified for a total of 13–17 cycles following transposition. Libraries were cleaned to remove very large (>1000) or very small (<50 bp) DNA sequences using AMPure bead cleanup. Library quality was assessed by Bioanalyzer.

## RNA-Seq

For RNASeq, ~25 ng of total RNA was using the NuGEN Ovation V2 kit for cDNA amplification, followed by NEBNext Ultra DNA Library Prep Kit for Illumina for library construction, according to manufacturer's instructions. A Covaris E220 was used to sonicate cDNA to ~200 bp fragments prior to library construction. Library quality was assessed by Agilent Bioanalyzer. Using a NextSeq 550 sequencer, 75 bp single-end sequencing was performed. Libraries were aligned and quantified using STAR v2.7.0 (*Dobin et al., 2013*). The mouse reference genome was generated using GRCm38, and gene counts quantified using annotations from GRCm38.99. Parameters used were `–runThreadN 8`; `–genomeDir` GRCm38; `–sjdbGTFFile` GRCm38.99; `–sjdbOverHang` 100; `–outSAMtype` BAM SortedByCoordinate; `–quantMode` GeneCounts.

## ATACSeq

ATACSeq libraries were sequenced on a NextSeq 550 using 35 × 2 reads. Raw data was cleaned and trimmed using fastp (*Chen et al., 2018*), and mapped to the mm10 mouse genome using bowtie2, v 2.3.4 (*Langmead and Salzberg, 2012*). Maximum insert size was set to 2000 (-X 2000). Mapped reads were converted to bam and filtered for quality using samtools (*Li et al., 2009*), with the following parameters: *samtools view -h -F 4 -q 10 -bS*. Duplicates were removed with Picard (http://broadinstitute.github.io/picard/), using MarkDuplicates (-REMOVE_DUPLICATES True). Libraries were then downsampled to 50 million reads, and Genrich (https://github.com/jsh58/Genrich) was used to call peaks on non-mitochondrial reads using ATAC-seq mode. For peak calling, files were sorted by name using samtools, and blacklist regions were derived from ENCODE. Both replicates were used for consenses peak claing. Parameters for Genrich used were *-j -v -y -r -e chrM, chrY -E $BLACKLIST -k OUTFILE.bedgraphish -t REPLICATE1.BAM,REPLICATE2.BAM -o OUTFILE. narrowPeak.* For genome browser visualization, bigwig files (.bw) were generated by text parsing to convert. bedgraphish outputs to. bedgraph, and converted to bigwig format bedGraphtoBigWig (http://genome.ucsc.edu/).

Finally, differential peak analysis was performed using Diffbind (*Ross-Innes et al., 2012*) with peaks generated from Genrich. Deeptools (*Ramírez et al., 2016*) was used to generate heatmap and profile plots, and Homer (*Heinz et al., 2010*) for transcription factor motif analysis and peak annotation.

## Single-cell data analysis

In the mouse, in order to identify gene sets representative of endothelial, cardiomyocyte, or fibroblast cells within the heart, gene counts from SMART-Seq2 RNA-Seq were downloaded from *Tabula Muris* (https://figshare.com/articles/dataset/Single-cell_RNA-seq_data_from_Smart-seq2_sequencing_of_FACS_sorted_cells_v2_/5829687). Data was processed using Seurat v3. Cells were first filtered for number of unique genes (nFeature_RNA > 200,<3000), and log normalized (normalization.method="LogNormalize', scale.factor = 10000). Data were then clustered by non-linear dimensional reduction (UMAP/tSNE) to identify cell type, using published RDS file to preserve clusters. Differentially expressed features (genes) were identified using with FindMarkers. The top 300 genes that were positively associated as markers of each cluster (that is, had increased rather than decreased expression) were used to generate cell-type specific gene lists. Marker gene lists and according statistics for each cell subset are shown in *Supplementary file 1*. To create bed files to generate enrichment tracks for ATACSeq, mitochondrial genes were removed from cell-type specific gene sets, since these genes are accessible in all cell types. The characteristic genes for each major cardiac cell type (cardiomyocyte, endothelial, fibroblast) were used for to analyze and compare accessibility of cell-type specific gene sets.

## Velocyto

To quantify spliced and unspliced reads across cardiac cell populations, we analyzed BAM alignment files from *Tabula Muris* using Velocyto (*La Manno et al., 2018*). Cells within cardiomyocyte, endothelial, or fibroblast cell populations were selected using the published annotations in *Tabula Muris*. For higher coverage, we used the SMART-Seq2 data from the microwell experiments. Alignment was performed using STAR. Resulting BAM files were first processed using Velocyto.py with run_smart-seq2 settings to generate loom files for each population. Spliced and unspliced counts were extracted using the Veloctyo.R analysis pipeline, and summed on a per-gene basis across all cells within each population.

## Primary mouse EC isolation

ECs were isolated from CDH5-Cre/NuTRAP animals from heart or lung as previously described (*Sawada et al., 2008*). In brief, tissues were minced, and digested in 2 mg/mL of collagenase I in DMEM for 20 min at 37°C. Cell suspensions were filtered through 40 µM filtered, and incubated with CD31-conjugated magnetic dynabeads for 1 hr at 4°C. Beads were washed on a magnetic column 8–10 times, and bead-bound cells were grown on gelatin coated tissue culture plates. For ATACSeq of isolated ECs, cells isolated from one animal (heart or lung) were passaged two to four times and grown to confluency for at least 2 days before transposition.

## Immunohistochemistry and confocal imaging

For histology, hearts were prepared for paraffin embedding and sectioned. Samples were hybridized for *Tnnt2* and *Cdh5* RNAs using probes and target amplification by ACD (RNAScope Multiplex Florescent Assay v2 kit), and co-stained by for Pecam1 (Cd31) protein. Imaging was conducted by laser scanning confocal microscopy using a Zeiss LSM 710 Confocal microscope. Images were taken at 63X. Z-stacks were taken at 0.48 um apart. For representative images, three slices are merged and shown (final thickness 0.96 µm).

## Statistics

Pair-wise comparisons were analyzed in Prism by two-tailed unpaired Student's t-test, and p<0.05 was considered statistically significant. For sequencing analysis, statistics were performed using the programs outlined above.

## Acknowledgements

ZA is supported by NIH R01HLDK114103 and the AHA/Allen Foundation. NY was supported by NIH T32 DKO7314.

## Additional information

### Funding

| Funder | Grant reference number | Author |
|---|---|---|
| National Institute of Diabetes and Digestive and Kidney Diseases | R01HLDK114103 | Zoltan Arany |
| American Heart Association | AHA/Allen Initative | Zoltan Arany |
| National Institutes of Health | T32 DKO7314 | Nora Yucel |

The funders had no role in study design, data collection and interpretation, or the decision to submit the work for publication.

### Author contributions

Nora Yucel, Conceptualization, Formal analysis, Investigation, Visualization, Writing - original draft; Jessie Axsom, Data curation, Formal analysis, Investigation, Visualization, Writing - review and editing; Yifan Yang, Formal analysis, Investigation, Methodology; Li Li, Resources, Investigation, Methodology; Joshua H Rhoades, Formal analysis, Supervision, Investigation, Methodology, Writing - original draft; Zoltan Arany, Conceptualization, Data curation, Formal analysis, Supervision, Funding acquisition, Investigation, Methodology, Writing - original draft, Writing - review and editing

### Author ORCIDs

Nora Yucel (iD) https://orcid.org/0000-0002-1848-6462
Zoltan Arany (iD) https://orcid.org/0000-0003-1368-2453

### Ethics

Animal experimentation: This study was performed in strict accordance with the recommendations in the Guide for the Care and Use of Laboratory Animals of the National Institutes of Health. All of the animals were handled according to approved institutional animal care and use committee (IACUC) protocols (#805255) of the University of Pennsylvania.

### Decision letter and Author response

Decision letter https://doi.org/10.7554/eLife.55730.sa1
Author response https://doi.org/10.7554/eLife.55730.sa2

# Additional files

### Supplementary files

• Supplementary file 1. Markers of cardiomyocyte, endothelial cell, and fibroblast cell subsets in *Tabula Muris* heart data.

• Supplementary file 2. Velocyto splicing analysis of cardiomyocyte, endothelial cell, and fibroblast cell subsets in *Tabula Muris* heart expression data.

• Supplementary file 3. Diffbind analysis and annotation of ATACSeq data from GFP- and GFP+ nuclei subsets of NuTRAP hearts.

• Supplementary file 4. Diffbind analysis and annotation of ATACSeq data from freshly isolated GFP + endothelial cell nuclei vs isolated and cultured cardiac endothelial cells.

• Transparent reporting form

### Data availability

The endothelial translatome RNA-Seq data for Rpl22/Tek tissues from Cleuren, et al (2019) was extracted from Gene expression Omnibus (GEO) under accession number GSE138630. Microarray expression data of microvascular endothelial cells isolated by surface marker staining (Nolan et al,

2013) was retrieved from GSE47067. Microarray expression data from Coppiello, et al 2015, of Tie2-GFP labeled endothelial cells was obtained under GSE48209. The human fetal endothelial cell RNA-Seq data-set from Marcu, et al 2018 was obtained under accession GSE114607. Tabula Muris data is available under accession GSE109774, or on FigShare (https://figshare.com/projects/Tabula_Muris_Transcriptomic_characterization_of_20_organs_and_tissues_from_Mus_musculus_at_single_cell_resolution/27733).

The following dataset was generated:

| Author(s) | Year | Dataset title | Dataset URL | Database and Identifier |
|---|---|---|---|---|
| Yucel N, Axsom J, Yang Y, Li L, Rhoades JH, Arany Z | 2020 | Cardiac endothelial cells maintain open chromatin and expresion of cardiomyocyte myofibrillar genes | https://www.ncbi.nlm.nih.gov/geo/query/acc.cgi?acc=GSE144839 | NCBI Gene Expression Omnibus, GSE144839 |

The following previously published datasets were used:

| Author(s) | Year | Dataset title | Dataset URL | Database and Identifier |
|---|---|---|---|---|
| Cleuren ACA, van der Ent MA, Jiang H, Hunker KL | 2019 | The in vivo endothelial cell translatome is highly heterogeneous across vascular beds | https://www.ncbi.nlm.nih.gov/geo/query/acc.cgi?acc=GSE138630 | NCBI Gene Expression Omnibus, GSE138630 |
| Coppiello G, Collantes M, Sirerol-Piquer MS, Vandenwijngaert S | 2015 | Meox2/Tcf15 heterodimers program the heart capillary endothelium for cardiac fatty acid uptake | https://www.ncbi.nlm.nih.gov/geo/query/acc.cgi?acc=GSE48209 | NCBI Gene Expression Omnibus, GSE48209 |
| Nolan DJ, Ginsberg M, Israely E, Palikuqi B | 2013 | Molecular signatures of tissue-specific microvascular endothelial cell heterogeneity in organ maintenance and regeneration | https://www.ncbi.nlm.nih.gov/geo/query/acc.cgi?acc=GSE47067 | NCBI Gene Expression Omnibus, GSE47067 |
| Marcu R, Choi YJ, Xue J, Fortin CL | 2018 | Human Organ-Specific Endothelial Cell Heterogeneity | https://www.ncbi.nlm.nih.gov/geo/query/acc.cgi?acc=GSE114607 | NCBI Gene Expression Omnibus, GSE114607 |
| Tabula Muris Consortium | 2018 | Tabula Muris: Transcriptomic characterization of 20 organs and tissues from Mus musculus at single cell resolution | https://www.ncbi.nlm.nih.gov/geo/query/acc.cgi?acc=GSE109774 | NCBI Gene Expression Omnibus, GSE109774 |

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
