## [Decision Letter]

**Acceptance summary:**

Your demonstration that myofibrillar genes are transcribed in cardiac endothelial cells using epigenetic, RNAseq and RNAscope/imaging criteria are appropriate for a Research Advance.

**Decision letter after peer review:**

Thank you for submitting your article "Cardiac endothelial cells maintain open chromatin and expression of cardiomyocyte myofibrillar genes" for consideration by *eLife*. Your article has been reviewed by three peer reviewers, and the evaluation has been overseen by a Reviewing Editor and Anna Akhmanova as the Senior Editor. The reviewers have opted to remain anonymous.

The reviewers have discussed the reviews with one another and the Reviewing Editor has drafted this decision to help you prepare a revised submission.

The editors have judged that your manuscript is of interest and potentially valuable to the field, but as described below, additional experiments are required before it is considered for publication. We would like to draw your attention to changes in our revision policy that we have made in response to COVID-19 (https://elifesciences.org/articles/57162). First, because many researchers have temporarily lost access to the labs, we will give authors as much time as they need to submit revised manuscripts. We are also offering, if you choose, to post the manuscript to bioRxiv (if it is not already there) along with this decision letter and a formal designation that the manuscript is 'in revision at *eLife*'. Please let us know if you would like to pursue this option. (If your work is more suitable for medRxiv, you will need to post the preprint yourself, as the mechanisms for us to do so are still in development.)

Summary:

Yucel and colleagues propose that cardiac endothelial cells (EC) have an epigenetic and transcriptional profile that partly mirrors that of cardiomyocytes. Specifically, the authors report the expression of cardiomyocyte myofibril genes (e.g., troponin, titin), which are present in cardiac endothelial cells but not in endothelial cells from other organs. The chromatin of these genes is in an "open state" , which appears to be dependent on the cardiac micro-environment. The authors speculate that this tissue-specific gene expression signature may reflect the developmental origin of cardiac endothelial cells. This work is provocative, and as such reviewers felt that more rigorous work is needed to support the conclusions.

Essential revisions:

Essential revisions focus primarily on rigorous experimentation to more unambiguously show that the "cardiac gene expression signature" is indeed cell-intrinsic to EC. It was pointed out that the presence of cardiomyocyte RNAs in the EC single-cell and ATAC seq data might be due to contamination, and that more rigorous data is needed to show in situ endothelial cell expression of cardiomyocyte genes unambiguously. There are also numerous instances of vague descriptions of approaches, methods and analysis that need to be improved. Essential points:

1) Several reviewers were concerned with the possibility that cardiomyocyte RNAs could be represented in the single-cell datasets but not originate from endothelial cells. They suggested the possibility of cardiomyocyte-EC doublets, or cell-free RNA sticking to antibodies or beads, or to glycocalyx of the EC cells. They point out that the ATAC-seq data is on bulk RNA and so could be contaminated by some cardiomyocytes, and that only the most abundant cardiomyocyte RNAs seem to be expressed by EC and represented in the ATAC-seq data. It was difficult to evaluate the interpretation of the data because the negative population in the TRAP experiment is not clearly defined, and if whole heart tissue is open to contamination from cardiomyocytes. Therefore more rigorous experimentation is required, such as dual single-cell RNA and ATAC-seq to show that bona fide EC have epigenetic patterns suggestive of cardiomyocyte gene expression. It would also strengthen the data to do single-cell RNA seq analysis before and after RNAse treatment to evaluate potential capture of negatively charged RNA by positively charged EC glycocalyx. It was suggested that a better analysis of the epigenetic landscape of cardiac genes in EC would be informative – for example, the chromatin status of other cardiac genes not found in heart EC is of interest. What about epigenetic marks that indicate active transcription of genes? Are these marks found in genes like Tnni, Tnnt2, Tnnc1 an Myh6 when analyzed in cardiac endothelial cells? And why closed chromatin is found in active genes (Figure 5) is not well-explained.

2) The RNA scope data was insufficient to determine whether the signals only originate from EC. This needs to be higher resolution and accompanied by antibody staining of EC-specific junction markers to clearly show EC cell borders. It was also suggested that antibodies to the cardiac proteins expressed by EC should be used with EC markers and high-resolution imaging, since the RNAs appear to be abundant (Figure 1).

3) There are numerous places where the methodology is vague, and where approaches and analysis are poorly described or documented.

[Editors' note: further revisions were suggested prior to acceptance, as described below.]

Thank you for resubmitting your work entitled "Cardiac endothelial cells maintain open chromatin and expression of cardiomyocyte myofibrillar genes" for further consideration by *eLife* as a Research Advance. Your revised article has been reviewed by two peer reviewers and the evaluation has been overseen by Anna Akhmanova as the Senior Editor and a Reviewing Editor.

The manuscript has been significantly improved, but there are some remaining issues that need to be addressed before acceptance, as outlined below:

1) The RNA scope data is much improved, but some concerns regarding specificity remain. Thus we request a Z-plane view of the CDH5/TNNT2 overlap from the RNA scope data – to document that the overlap extends to the Z plane. This should be doable from the images at hand.

2) We ask that authors amend the Discussion to better reflect the idea that the work focuses on transcriptional and epigenetic regulation of myocardial genes in cardiac endothelial cells, and that protein translation/function remains an open question.

---

## [Author Response]

Essential revisions:Essential revisions focus primarily on rigorous experimentation to more unambiguously show that the "cardiac gene expression signature" is indeed cell-intrinsic to EC. It was pointed out that the presence of cardiomyocyte RNAs in the EC single-cell and ATAC seq data might be due to contamination, and that more rigorous data is needed to show in situ endothelial cell expression of cardiomyocyte genes unambiguously. There are also numerous instances of vague descriptions of approaches, methods and analysis that need to be improved. Essential points:1) Several reviewers were concerned with the possibility that cardiomyocyte RNAs could be represented in the single-cell datasets but not originate from endothelial cells. They suggested the possibility of cardiomyocyte-EC doublets, or cell-free RNA sticking to antibodies or beads, or to glycocalyx of the EC cells. They point out that the ATAC-seq data is on bulk RNA and so could be contaminated by some cardiomyocytes, and that only the most abundant cardiomyocyte RNAs seem to be expressed by EC and represented in the ATAC-seq data. It was difficult to evaluate the interpretation of the data because the negative population in the TRAP experiment is not clearly defined, and if whole heart tissue is open to contamination from cardiomyocytes. Therefore more rigorous experimentation is required, such as dual single-cell RNA and ATAC-seq to show that bona fide EC have epigenetic patterns suggestive of cardiomyocyte gene expression. It would also strengthen the data to do single-cell RNA seq analysis before and after RNAse treatment to evaluate potential capture of negatively charged RNA by positively charged EC glycocalyx. It was suggested that a better analysis of the epigenetic landscape of cardiac genes in EC would be informative – for example, the chromatin status of other cardiac genes not found in heart EC is of interest. What about epigenetic marks that indicate active transcription of genes? Are these marks found in genes like Tnni, Tnnt2, Tnnc1 an Myh6 when analyzed in cardiac endothelial cells? And why closed chromatin is found in active genes (Figure 5) is not well-explained.

We thank the reviewers for their critical evaluation of this important question of whether cross contamination between cardiomyocytes and ECs could explain the presence of myofibril gene RNAs in ECs. We, too, were very concerned about this possibility at first. We wish to clarify our ATAC-Seq data and RNAseq data, and better articulate why such cross-contamination cannot explain our findings:

1) “ATAC-seq data is on bulk RNA and so could be contaminated by some cardiomyocytes”: the ATAC-seq studies were performed on DNA, not RNA. In addition, for ATAC-Seq, it is important to note that the Tn5 transposase is specific for double stranded DNA, and thus will not be affected by cell-free RNA.

2) “Cardiomyocyte-EC doublets”: at the level of cells (i.e. Tabula Muris data) this is highly unlikely, as few cardiomyocytes were sorted in this dataset, due to their large size. At the level of nuclei sorting (our data), endothelial cell nuclei were sorted on the basis of DAPI staining, and doublets were excluded during sorting by both DAPI staining and FSC-A/FSC-H gating by flow cytometry. This is shown in a newly added Figure 3—figure supplement 1. We also assessed the exclusion of cardiomyocyte nuclei by co-staining of nuclei for PCM1, a specific marker of cardiomyocyte nuclei, and find no PCM1 signal in our EC nuclei fraction (Figure 3A). It is thus highly unlikely that doublets were included, and even if so, the numbers would be low.

3) “Only the most abundant cardiomyocyte RNAs seems to be expressed by EC and represented in the ATAC-seq data”. It is important to note that the ATAC-seq data evaluate chromatin, not RNA expression. Thus, contamination would have to occur at the level of chromatin DNA, not RNA, and the level of expression in cardiomyocytes would have no bearing on the ATAC-seq data. So, for example, if 10% of EC nuclear prep was contaminated by cardiomyocyte nuclei, then we would detect 10% open chromatin at the troponin genome locus, no matter the level of expression of troponin. Instead, we find 100% open chromatin at the troponin locus in the EC prep (Figure 4A). Contamination by CM nuclei cannot explain this observation.

4) “Cell-free RNA sticking to antibodies or beads, or to glycocalyx of the EC cells.” We do not exclude the contribution of contamination from cardiomyocyte RNA, which is inevitable in any RNA-Seq study. However, we performed analyses to demonstrate that such contamination, to the extent that it does occur, is minimal. First, we find that not all abundant cardiomyocyte RNAs are expressed in ECs, while, if contamination was indeed occurring, then all abundant RNAs should be detected, in relative proportion to their abundance. To quantify this observation, we followed the same approach as Jambusaria, et al., 2020, and performed Kendall-Tau rank-score statistical analysis on the highest 300 expressed genes in EC vs. non-EC material. We performed this comparison on both our own TRAP data, as well as TRAP data recently published in Cleuren, et al., 2019. In both our own data, as well as that from Cleuren, et al., there was a low correlation (<0.25), indicating that the ordering of highly expressed genes, of which many are CMF genes, is only poorly correlated in EC vs. non-EC samples. We clarify this important point in the text and new Figure 1B.

5) “Cell-free RNA sticking to antibodies or beads, or to glycocalyx of the EC cells.” The single cell RNAseq datasets demonstrate this not to be the case, as *only a subset* of ECs contained *only a subset* of CMF genes (most often only one) (Figure 2B). If CMF mRNAs were being captured by ECs (e.g. by positively charged EC glycocalyx), this would be observed in all cells; and even if only a subset of ECs captured CMF mRNAs (e.g. if only some ECs remained covered with glycocalyx) they would capture *all* CMF genes instead of on average only one. In addition, a glycocalyx explanation cannot explain the observations made with ribosome pulldowns, as seen in our data and that of Jambusaria et al., and that of Cleuren et al.

6) “The chromatin status of other cardiac genes not found in heart EC is of interest”. We agree. Strikingly, the majority of cardiomyocyte-specific genes reveal open chromatin in ECs (new Figure 4—figure supplement 1A) in particular in promoter regions (Figure 4C). Nevertheless, quantitatively, there are some cardiomyocyte-specific enhancer regions (intronic or intergenic) associated with CMF genes that are not found in endothelial cells (new Figure 4—figure supplement 1B). We show a few examples, including an enhancer region upstream of the gene encoding the cardiac transcription factor Nkx2-5, and an intronic enhancer within the Dmd gene (new Figure 4—figure supplement 1C). Once again, contamination cannot explain these data.

7) Finally, to provide yet more evidence that the expression of CMF genes in ECs is cell autonomous, we have quantified spliced vs. un-spliced reads, reasoning that if contamination were occurring, few un-spliced CMF mRNAs (i.e. nascent mRNAs) would be detected in ECs. Instead, we find relatively *more* un-spliced CMF mRNAs in ECs (new Figure 2D-E, Supplementary file 2), again supporting the notion that these mRNAs are expressed directly from open chromatin in the ECs themselves.

In summary, we provide extensive demonstration to address the fundamental question being posed of whether RNAs or DNAs originating from cardiomyocytes are “contaminating” the observed EC-specific RNAseq and ATACseq. We now articulate in the Discussion in more detail the numerous reasons why this cannot be the case, with the following paragraph:

“(1) Kendall Tau correlations demonstrate lack of correlation between genes highly expressed in cardiomyocytes and those potentially “contaminating” ECs. Were contamination to be occurring, these levels of expression would correlate. […] Thus, if contamination were occurring in this setting, EC chromatin would be, on average, only mildly open (proportional to the level of contamination), because the majority of the signal would still stem from ECs.”

2) The RNA scope data was insufficient to determine whether the signals only originate from EC. This needs to be higher resolution and accompanied by antibody staining of EC-specific junction markers to clearly show EC cell borders. It was also suggested that antibodies to the cardiac proteins expressed by EC should be used with EC markers and high-resolution imaging, since the RNAs appear to be abundant (Figure 1).

We provide new RNAscope data, as suggested, including higher resolution images, and accompanied by antibody staining of EC-specific junction maker Pecam1 (new Figure 3D). Additional images are also provided in Figure 3—figure supplement 2. We find that the proportion of ECs with detectable nuclear Tnnt2 mRNA on RNAscope is nearly identical to the proportion of ECs in the Tabular Muris data that express Tnnt2 (Quantification now moved to Figure 3—figure supplement 2D).

3) There are numerous places where the methodology is vague, and where approaches and analysis are poorly described or documented.

Methodologies and explanation of analyses have been expanded to be made more clear throughout.

[Editors' note: further revisions were suggested prior to acceptance, as described below.]

The manuscript has been significantly improved, but there are some remaining issues that need to be addressed before acceptance, as outlined below:1) The RNA scope data is much improved, but some concerns regarding specificity remain. Thus we request a Z-plane view of the CDH5/TNNT2 overlap from the RNA scope data – to document that the overlap extends to the Z plane. This should be doable from the images at hand.

We have added X and Y views to the RNAscope images in order to better show overlap in the CD31+ regions.

2) We ask that authors amend the Discussion to better reflect the idea that the work focuses on transcriptional and epigenetic regulation of myocardial genes in cardiac endothelial cells, and that protein translation/function remains an open question.

We have added lines in the Discussion to emphasize that our work focuses on epigenetics and transcription, and that translation and function are questions that remain to be answered.